# Frugoside Induces Mitochondria-Mediated Apoptotic Cell Death through Inhibition of Sulfiredoxin Expression in Melanoma Cells

**DOI:** 10.3390/cancers11060854

**Published:** 2019-06-19

**Authors:** In-Sung Song, Yu Jeong Jeong, Ji Eun Kim, Jimin Shin, Sung-Wuk Jang

**Affiliations:** 1Department of Biomedical Sciences, Asan Medical Center, University of Ulsan College of Medicine, Seoul 138-736, Korea; microvirus@ulsan.ac.kr (I.-S.S.); jyj2479@ulsan.ac.kr (Y.J.J.); stacy1210@yonsei.ac.kr (J.E.K.); jimin9966@ulsan.ac.kr (J.S.); 2Asan Medical Institute of Convergence Science and Technology, Asan Medical Center, University of Ulsan College of Medicine, Seoul 138-736, Korea; 3Department of Biochemistry and Molecular Biology, University of Ulsan College of Medicine, Seoul 138-736, Korea

**Keywords:** frugoside, peroxiredoxin, sulfiredoxin, reactive oxygen species, melanoma

## Abstract

Malignant melanoma is the most life-threatening neoplasm of the skin. Despite the increase in incidence, melanoma is becoming more resistant to current therapeutic agents. The bioactive compound frugoside has been recently reported to inhibit growth when used in various cancer cells. However, this effect has not been demonstrated in melanoma. Here, we found that frugoside inhibited the rate of reduction of hyperoxidized peroxiredoxins (Prxs) by downregulating sulfiredoxin (Srx) expression. Furthermore, frugoside increased the accumulation of sulfinic Prxs and reactive oxygen species (ROS) and stimulated p-p38 activation, resulting in the mitochondria-mediated death of M14 and A375 human melanoma cells. The mitochondria-mediated cell death induced by frugoside was inhibited by the overexpression of Srx and antioxidants, such as N-acetyl cysteine and diphenyleneiodonium. In addition, we observed that frugoside inhibited tumor growth without toxicity through a M14 xenograft animal model. Taken together, our findings reveal that frugoside exhibits a novel antitumor effect based on a ROS-mediated cell death in melanoma cells, which may have therapeutic implications.

## 1. Introduction

Melanoma, a type of skin cancer, exhibits traits that include high metastatic potential, resistance to chemotherapy, and a poor survival rate [1,2,3]. According to epidemiological studies, the incidence of melanoma increases at a faster rate than any other cancer worldwide [4,5]. Recently, many chemo- and immune-based therapies or targeted therapies have been evaluated for clinical trials [6,7,8,9,10]. However, as most conventional therapies have failed to show a significant benefit in melanoma patients, there is an urgent need to find adequate agents with a lower toxicity to treat malignant melanomas. We thus require novel therapeutics with a higher efficiency and fewer side effects; therefore, natural compounds might play a crucial role in the development of new anti-cancer agents [11,12,13]. Natural compounds have long been known as sources of drugs against several human diseases, including cancer. Vincristine, etoposide, and paclitaxel are examples of pharmaceuticals derived from plants [14]. Despite the discovery of natural cytotoxic agents, the search for new anti-cancer drugs remains necessary in order to meet the demand for fewer severe side effects and higher efficiency in therapies.

The bioactive compound frugoside, a new cardenolide glycoside from the leaves of *Calotropis procera* W.T. Aiton (family Asclepiadaceae), was recently reported to inhibit the growth of various human cancer cells, including non-small cell lung cancer, glioblastoma, and prostate cancer cell lines [15,16,17,18]. However, the biological effect of frugoside on melanoma cells has not been evaluated.

Reactive oxygen species (ROS), known as secondary messengers in intracellular signaling, contribute to cancer progression and development at low levels; however, at high levels, ROS can act as an anti-tumor species by inducing cell senescence and apoptosis. In fact, cancer cells do exhibit an abnormal redox status followed by increased basal ROS production, and they thus cannot tolerate higher levels of free radicals [19]. Indeed, recent studies have shown that ROS, generated through redox dysregulation, contribute to the malignant transformation and progression of melanoma by altering cellular signaling and survival pathways [19]. Therefore, a compound that hinders redox regulation and selectively targets tumors might be a promising treatment, especially for melanoma.

Recent data showed that antioxidant proteins protect several types of cancer cells from oxidative stress. The enzymes include catalase, glutathione peroxidase (GPx), and peroxiredoxins (Prxs). The predominant enzymes responsible for the elimination of H_2_O_2_ in cells are Prxs and catalase. Catalase is exclusively localized in peroxisomes and eliminates H_2_O_2_ when it is present at a high concentration compared with other antioxidant proteins. Kinetic and structural analyses have revealed that Prxs possess an active-site pocket that gives rise to a high-affinity peroxide binding site that is absent in catalase and GPxs [20,21,22]. As a consequence, Prxs are the major cellular antioxidants that scavenge peroxides and mediate H_2_O_2_-induced intracellular signaling. Prxs comprise three subfamilies: 2-Cys (PrxI to PrxIV), atypical 2-Cys, and 1-Cys [23,24]. The 2-Cys Prxs exist as homodimers and contain two conserved cysteine residues. The N-terminal Cys-SH is first oxidized by peroxides to Cys-SOH, and it then forms a disulfide bond together with the C-terminal Cys-SH of the other subunits. This disulfide is specifically reduced by thioredoxin, whose oxidized thioredoxin is then reduced by thioredoxin reductase. The sulfenic intermediates (Cys-SOH) are occasionally further oxidized to cysteine sulfinic acid (Cys-SO_2_H), which causes the inactivation of peroxidase that cannot be reduced by thioredoxin [25].

Sulfiredoxin (Srx) is an important enzyme that protects against oxidative damage of host cells through the reduction of hyperoxidized peroxiredoxin (Prx-SO_2_H), a type of cellular antioxidant [26,27,28]. However, the importance of Srx in the pathogenesis of human diseases, including cancer, is poorly understood. Recent reports indicate that Srx is overexpressed in a variety of cancers and may promote tumorigenesis in a Prx-dependent or independent manner [26,27,28]. It is therefore important to address Srx regulation.

In the present study, we reported that frugoside induces oxidative mitochondrial damage and mitochondria-mediated apoptotic cell death by inhibiting Srx expression and delaying the reduction of hyperoxidized Prx in melanoma cells. Our results suggest that frugoside might be a potential therapeutic agent for melanoma treatment.

## 2. Results

### 2.1. Frugoside Leads to Attenuated Srx Expression and Subsequently Delays Reduction of Hyperoxidized Prxs in Melanoma Cells

Srx is crucial for cellular redox homeostasis and cancer progression. Additionally, redox dysregulation is very important for malignant transformation and progression in melanoma. Therefore, we first examined the expression of Srx in various melanoma cells. As shown in Appendix A, Srx was highly expressed in melanoma cells. From these data and recent reports [26], we verified Srx as a drug target to develop anti-cancer drug treatments against melanoma. One hundred compounds screened from the in-house library using the western blot assay with the Prxs-SO_2_ antibody to determine Srx inhibitors. The screening identified the inhibitor frugoside and its chemical structure (Appendix A). To confirm the influence of frugoside, we examined its dose- and time-dependent effects on Srx expression. As shown in Figure 1A,B, the expression of Srx decreased in time- and dose-dependent manners in M14 and A375 human melanoma cells. In contrast, other antioxidant proteins, such as Prx2 and Prx3, were not decreased by frugoside treatment. To further confirm whether decreased Srx expression affects hyperoxidized Prxs, we examined the reduction of sulfinic Prxs in melanoma cells cultured in the presence of frugoside. Consequently, the decreased Srx expression by frugoside treatment resulted in the attenuation of the reducing rate of sulfinic Prxs in melanoma cells (Figure 1C,D). The inhibition of Srx protein expression did not contribute to the reduction of peroxidized Prx, which resulted in the intracellular accumulation of ROS [19]. Frugoside treatment significantly increased intracellular ROS in dose- and time-dependent manners (Figure 1E,F and Appendix A). Taken together, these results suggest that frugoside inhibits Srx expression and may lead to cell death via the accumulation of intracellular ROS in melanoma cells.

### 2.2. Frugoside Induces Mitochondria-Mediated Apoptotic Cell Death in Melanoma Cells

We examined whether frugoside induced cell death due to the accumulation of intracellular ROS in melanoma cells. Frugoside treatment induced the death of melanoma cells in time- and dose-dependent manners (Figure 2A,B and Appendix A). Cell death was also confirmed using a fluorescence-activated cell sorting analysis after Annexin V-fluorescein isothiocyanate (V-FITC) and propidium iodide (PI) staining in frugoside-treated M14 and A375 melanoma cells. The frugoside-induced cell death occurred in a dose-dependent manner (Figure 2C,D). To verify whether the cell death induced by frugoside was due to apoptosis, the sub-G1 phase DNA content of the cell cycle was measured in frugoside-treated melanoma M14 and A375 cells in dose-dependent manners. Frugoside increased the sub-G1 population, which is an indication of increased apoptosis, in a dose-dependent manner (Figure 2E,F). Next, an immunoblot analysis was used to determine whether frugoside treatment altered part of the cell death cascade, such as caspase-3 and poly (adenosine diphosphate (ADP)-ribose) polymerase (PARP), in a dose- or time-dependent manner. The levels of cleaved caspase-3 and PARP increased in a dose- and time-responsive manner in frugoside-treated melanoma M14, A375 (Figure 2G,H) and A2068 cells (Appendix A). Lastly, to determine whether mitochondria were involved in the frugoside-induced cell death, we analyzed the release of cytochrome c, the apoptotic signal mediator. Cytochrome c markedly increased in the cytosol according to the frugoside dose (Figure 2I,J and Appendix A). Furthermore, Bcl2 expression, which protects cells against diverse cytotoxic signals that trigger the mitochondrial apoptotic pathway by preventing mitochondrial alterations and cytochrome c release, was dramatically decreased in response to frugoside treatment in both M14 and A375 cells (Figure 2K and Appendix A). Collectively, these results indicate that frugoside presumably induces mitochondria-mediated apoptotic cell death via the inhibition of Srx expression. Moreover, mitochondria may function as apoptosis regulators by directly activating the caspase cascade in frugoside-treated melanoma cells.

### 2.3. Frugoside Induces Oxidative Mitochondrial Damage

We demonstrated that frugoside led to the accumulation of intracellular ROS by inhibiting Srx protein expression, resulting in cell death. ROS promotes apoptosis by opening the mitochondrial permeability transition pore (MPTP) complex [29]. To test whether frugoside affects the depolarization of the mitochondrial membrane potential in melanoma cells, we pretreated the melanoma cells with frugoside for 24 h before treatment with tetraethylrhodamine ethyl ester (TMRE), a fluorescent dye that accumulates in mitochondria, to assess apoptosis or mitochondrial depolarization. Frugoside caused a dose- and time-dependent decrease in TMRE fluorescence (Figure 3A,B and Appendix A). Melanoma cells exposed to 0.5 μg/mL of frugoside for 24 h exhibited a greater than 50% decrease in mitochondrial membrane potential. Furthermore, we demonstrated that frugoside altered mitochondrial ROS, as measured by a FACS analysis using MitoSOX, a fluorogenic dye for the highly selective detection of superoxides in the mitochondria of live cells. Frugoside induced significant levels of superoxides in both dose- and time-dependent manners (Figure 3C,D and Appendix A). We also observed a calcium overload in frugoside-exposed cells, but this was not dependent on dose or time (Figure 3E,F and Appendix A). Finally, we confirmed that frugoside altered mitochondrial function using fluorescence microscopy (Figure 3G–I). Collectively, these data suggest that the Srx inhibitor frugoside induces the prompt accumulation of ROS in the cytosol (until 12 h), and that the accumulated ROS subsequently leads to the collapse of the mitochondrial membrane potential followed by the opening of the MPTP. The collapse of the mitochondrial membrane potential boosted ROS accumulation at late stages (24 h), leading to cell death together with the release of apoptotic proteins, including cytochrome c.

### 2.4. Frugoside Induces Cell Death via p38 MAPK Activation and ROS Accumulation

Several recent reports demonstrated that increased ROS levels activate c-Jun N-terminal kinases (JNKs), p38, or extracellular signal-regulated kinases (ERKs) according to the stimuli or cell type [30,31]. To explore whether frugoside possesses a similar increased ROS-dependent signaling cascade, we treated M14 and A375 cells with various doses of frugoside and for various lengths of time. As shown in Figure 4A and Appendix A, frugoside treatment induced the sustained activation of p38 mitogen-activated protein kinase (MAPK) and ERKs, but not JNK, based on dose and time in melanoma M14 and A375 cells. Frugoside treatment slightly increased the phosphorylation of ERKs compared to p38 MAPK. We next assessed whether the sustained phosphorylation of p38 MAPK and ERKs could influence cell death in response to frugoside treatment. To confirm the effects of p38 and ERKs in frugoside-treated melanoma cells, we used SB202190 and U0126, inhibitors of p38 MAPK and the MAP/ERK kinase (MEK) 1/2, respectively. When M14 cells were pretreated with the p38 inhibitor, frugoside-induced cell death decreased by approximately 30%. Additionally, the MEK1/2 inhibitor slightly diminished frugoside-induced cell death (Figure 4B). We also confirmed that p38 inhibitor pretreatment attenuated the frugoside-induced cleavage of caspase-3 and PARP protein in melanoma M14 cells (Figure 4C). However, frugoside-induced cell death was not completely mitigated by the p38 inhibitor, as shown in Figure 4B,C. Therefore, frugoside treatment induced p38 kinase phosphorylation due to ROS accumulation, which may have partially contributed to cell death.

To confirm that frugoside-mediated cytotoxic effects are due to ROS accumulation caused by Srx inhibition, we examined the effects of antioxidants on mitochondrial function, p38 MAPK phosphorylation, and cell death in melanoma cells exposed to frugoside. When the antioxidants N-acetyl cysteine (NAC) and diphenyleneiodonium (DPI) were used to pre-treat melanoma M14 and A375 cells prior to frugoside treatment, frugoside-induced cell death was significantly reduced to the basal level, as measured by a FACS analysis after staining with the Annexin V-FITC/PI reagent (Figure 4D and Appendix A). In addition, frugoside-mediated caspase-3 cleavage, which indicates apoptotic cell death, was decreased by pretreatment with NAC or DPI (Figure 4E). The depolarization of the mitochondrial membrane potential, indicating mitochondrial dysfunction and cell death, was also inhibited by pretreatment with NAC or DPI (Figure 4F and Appendix A). Finally, we showed that antioxidant pretreatment resulted in a decrease of sustained p38 MAPK activation due to ROS accumulation in frugoside-treated cells (Figure 4G). These data suggest that frugoside-mediated intracellular ROS causes mitochondrial dysfunction and the sustained activation of p38 MAPK, leading to mitochondria-mediated cell death.

### 2.5. Frugoside-Mediated Cytotoxic Effects Are Reduced by Srx Overexpression

To further explore whether this frugoside-mediated cytotoxic effect is required to affect Srx expression levels, we transfected M14 and A375 cells with HA-Srx. As shown in Figure 5A and Appendix A, the hyperoxidation of Prxs by H_2_O_2_ in frugoside-pretreated cells increased more than in vehicle-treated cells. Conversely, hyperoxidized Prxs were reduced when HA-tagged Srx was expressed in melanoma cells. The expression of the hyperoxidized Prxs, Prx2, and Prx3 was not altered in frugoside-treated and Srx-overexpressed cells. Additionally, the ectopic expression of Srx significantly reduced frugoside-induced cell death in melanoma M14 and A375 cells (Figure 5B,C and Appendix A). We next examined whether Srx overexpression attenuated apoptotic frugoside-induced cell death. Caspase-3 protein cleavage was decreased in Srx-overexpressed M14 cells compared to Mock-transfected cells (Figure 5D). Moreover, the phosphorylation of p38 MAPK due to frugoside-induced ROS was decreased in Srx-overexpressed M14 cells. Finally, we confirmed whether Srx overexpression influenced mitochondrial Prx3 oxidation via the subcellular fractionation of M14 cells. When treated with frugoside, cytosolic Prx oxidation was increased by H_2_O_2_ treatment, and the increase of cytosolic Prx oxidation was decreased by Srx overexpression. Mitochondrial Prx oxidation was also decreased by Srx overexpression (Figure 5E). Taken together, these data suggest that frugoside induces ROS accumulation via Srx inhibition, and the accumulated ROS results in cytosolic/mitochondrial Prx hyperoxidation and p38 MAPK activation, leading to cell death in melanoma cells.

### 2.6. Frugoside-Mediated Srx Deficiency Impairs Tumor Growth In Vivo

To further determine potential anti-tumorigenesis in vivo, the influence of frugoside on anchorage-independent growth was assessed using clonogenic and soft agar assays. As expected, frugoside significantly decreased colony formation compared with vehicle-treated M14 and A375 cells (Figure 6A). Similar observations were obtained in the soft agar assay in response to frugoside treatment (Figure 6B). Next, to measure the in vivo effects of frugoside, a mouse xenograft model was established by the subcutaneous injection of melanoma M14 and A375 cells. Mice injected with frugoside displayed a marked reduction in tumor volume compared with that of the controls (Figure 6C,D). Moreover, no significant changes in body weight or adverse effects were observed in the frugoside-treated mice (Appendix A). Collectively, our data suggest that frugoside treatment results in the reduction of tumorigenic ability in vitro and in vivo through intracellular ROS accumulation by inhibiting Srx protein expression and Prx hyperoxidation in melanoma cells (Figure 6E).

## 3. Discussion

Surgery, radiotherapy, chemotherapy, immunotherapy, and various biological agents are currently used to treat malignant melanoma. Despite this arsenal of treatments, the prognosis is poor [32,33]. Here, we demonstrated that frugoside robustly inhibits Srx expression. Frugoside is a new cardenolide glycoside extracted from the Asclepiadaceae family of plants. It has potent cytotoxic activity against various cancer cell lines [15,16,17,18]. Though the possible use of frugoside as an anti-cancer drug has been mentioned, the molecular mechanism of its anti-cancer function remains to be established. 

In the present study, we demonstrated that frugoside: (a) Inhibited Srx expression and subsequently delayed the reduction of hyperoxidized Prxs in melanoma cells, (b) regulated mitochondrial function, (c) induced ROS generation and p38 MAPK activation, and (d) consequently induced mitochondria-mediated apoptotic cell death in melanoma cells. 

It has been reported that high concentrations of ROS induced by metabolic abnormalities and oncogenic signaling play an important role in the development and progression of cancer cells, which are very dependent on the capacity of antioxidants to protect from oxidative stress caused by elevated ROS [34,35,36]. Recently, it was shown that some common antioxidants increase the proliferation of severe malignant melanoma [37]. Based on these results, many researchers have conducted various studies designed to inhibit ROS removal or identify agents that can increase ROS production to kill cancer cells [34,35,36].

Our results show that frugoside can increase intracellular and mitochondrial ROS by inhibiting Srx expression in M14, A375, and A2058 melanoma cells (Figure 1 and Figure 3 and Appendix A). Additionally, based on results indicating that frugoside-induced mitochondrial damage and apoptotic effects can be inhibited by antioxidants, such as NAC and DPI, it is presumed that frugoside can induce cancer cell death through an ROS-mediated mechanism (Figure 4).

The most common preclinical model among the in vivo models for anti-cancer testing is the murine model. A number of melanoma murine models have been used, including xenograft, syngeneic, and genetically engineered models (GEMs) [38]. Among these murine modes, GEMs use transgenic mice with altered gene expression to determine the mechanism of melanomagenesis (CDKN2A, RAS, BRAF). In contrast, a mouse human tumor xenograft model has been used to evaluate target therapies and to test the combination efficacy of therapeutic agents [39]. We also showed the effect of frugoside using transplantation-based models. Using the mice models demonstrated that intraperitoneally injected frugoside strongly inhibited tumor volume, indicating a chemosensitizing effect in vivo (Figure 6 and Appendix A)

Srx is an important enzyme that reacts with ROS metabolism and oxidative stress in mitochondria by reducing hyperoxidized Prxs in yeast, plants, and mammals [34,35,36]. Elevated Srx expression has been identified in a wide range of human cancers, including breast, colorectal, lung, and prostate cancers [40]. In addition, Srx is highly expressed in malignant human skin tumors, including melanoma [26,41]. Consistent with these observations, we found that Srx is highly expressed in melanoma cells compared to normal skin cells (Appendix A).

Recently, it was reported that the regulation of the expression of Srx is involved in cell growth due to the influence on the reduction of hyperoxidized Prxs [34,35,36]. Our data provide support for these results. We found that frugoside inhibited Srx expression and attenuated the rate of reduction of sulfinic Prxs in M14 and A375 human malignant melanoma cells (Figure 1C,D). Additionally, the overexpression of Srx resulted in a decrease in the level of the sulfinic form of Prxs in frugoside-treated cells. The decreased accumulation of sulfinic Prxs resulted in a decreased apoptosis rate in frugoside-treated cells (Figure 5).

The reduction of Srx by frugoside treatment resulted in the accumulation of ROS, leading to the activation of the serine/threonine kinase p38MAPK (Figure 4). Several studies reported that the overproduction of ROS results in the activation of MAP kinases, such as JNK, p38MAPK, and ERK, contributing to cell death [28]. H_2_O_2_ acts on apoptosis signal-regulating kinase 1 (ASK1), a mitogen-activated protein kinase kinase kinase (MAPKKK), to activate MAPK signaling [42]. H_2_O_2_ elicits ASK1 oligomerization, resulting in the autophosphorylation and activation of ASK1 [43,44]. ASK1 is important for the activation of p38MAPK and JNK. We suggest that the frugoside-induced cell death is maybe caused by ASK1-p38MAPK activation due to ROS overproduction. Actually, our data showed that the frugoside-induced cell death and p38MAPK activation was decreased by the pretreatment with the antioxidants NAC and DPI (Figure 4). Moreover, pretreatment with the p38MAPK inhibitor, SB202190, resulted in decreased frugoside-induced cell death. The collective data indicate that frugoside has excellent translational potential as a novel anti-cancer drug for the treatment of melanoma.

## 4. Materials and Methods

### 4.1. Cell Culture, Antibodies, and Chemicals

The human melanoma cell lines M14, A375, A2058, and LOXIMVI were obtained from the American Type Culture Collection (ATCC; Manassas, VA, USA). The cells were cultured in RPMI-1640 supplemented with 10% fetal bovine serum (FBS) at 37 °C in a humidified incubator containing 5% CO_2_. Normal human skin cells (Detroit 551) were obtained from ATCC and maintained in dulbecco modified eagle medium (DMEM) high glucose (Lonza, Basel, Switzerland) supplemented with 10% FBS. Frugoside was obtained from NPBANK (Gungsan, Korea). Antibodies against poly-(adenosine diphosphate-ribose) polymerase (PARP), caspase-3, phospho(p)-JNK (Thr183/Tyr185), p-p38 (Thr180/Tyr182), p-ERK (Thr202/Tyr204), JNK, p38, and ERK were purchased from Cell Signaling Technology (Danvers, MA, USA). Antibodies against Prx2, Prx3, Prx-SO2, and hemagglutinin epitope (HA) were purchased from Abfrontier (Seoul, Korea). Antibodies against Srx and tubulin were purchased from Santa Cruz Biotechnology (Santa Cruz, CA, USA). Antibodies to cytochrome c were purchased from BD Pharmingen (San Jose, CA, USA). SB202190, N-acetyl-L-cysteine (NAC), diphenyleneiodonium chloride (DPI), U0126, and hydrogen peroxide solution were purchased from Sigma (St Louis, MO, USA).

### 4.2. Extraction and Isolation of Frugoside

*Cardiospermum halicacadum* was purchased from the Kyungdong traditional herbal market, Seoul, Korea, in 2010. The dried whole plants of *C. halicacadum* (5 kg) were extracted with 80% MeOH (25 L) for 3 times at room temperature. The extract was filtered and concentrated using a rotary vacuum drier to give the MeOH extract. The MeOH extract (1.0 kg) was suspended with 2.5 L of H_2_O and the same volume of ethylacetate (EA). The EA soluble fraction (110 g) was separated into twenty-one fractions (CHM 1-21) by chromatography on a silica gel column eluted with a gradient of CHCl_3_ and MeOH (10:1 to 2:1). The fraction CHM 6 (1.2 g) was isolated by a octadecyl-silica column (MeOH:H_2_O = 10:1) to obtain nine fractions (CHM 6-1 ~ CHM 6-9). The subfractions of CHM 6-3 were purified by silica gel column eluted with an isocratic condition of CHCl_3_ and MeOH (5:1) to obtain frugoside (87 mg). The structure of frugoside was determined by a comparison of its spectroscopic data with those reported in the literature [45].

### 4.3. In Vitro Cell Death Assays

Melanoma cells were treated with frugoside at the doses or times indicated in the figures. Similarly, for the inhibitor experiments, melanoma cells were cultured in the presence or absence of NAC, DPI, or SB202190 to confirm the effect of frugoside. Cell death was measured by a FACS analysis after staining using the Annexin V-fluorescein isothiocyanate/propidium iodide (Annexin V-FITC/PI) staining kit (Roche, Nutley, NJ, USA). Cell viability was determined using a CCK-8 assay kit (Dojindo, Tokyo, Japan), according to the manufacturer’s instructions. Briefly, melanoma cells were inoculated into a 96-well microplate, and a CCK-8 solution (10 μL/100 μL medium) was added to each well. After incubation for 1–4 h in a CO_2_ incubator at 37 °C, the absorbance of each well was measured at 450 nm using a microplate reader (Molecular Devices, Sunnyvale, CA, USA) with a reference wavelength of 650 nm. Cell cycle distribution was determined by DNA staining with PI (Sigma). Cells were harvested and fixed in 70% ethanol. Cell pellets were suspended in PI and simultaneously treated with RNase at 37 °C for 30 min. The percentage of cells in different cell cycle phases was measured using a FACSCanto II flow cytometer (BD Biosciences, Franklin Lakes, NJ, USA).

### 4.4. Quantitative Reverse Transcriptase Polymerase Chain Reaction

RNA was extracted from the cells using an RNeasy Mini kit (Qiagen, Hilden, Germany). Briefly, 1.5 μg RNA was reverse-transcribed with oligo (dT) 12–18 primers using the First-Strand cDNA Synthesis Kit (Fermentas, Grand Island, NY, USA). All reactions were performed in triplicate, and the *B2M* gene was used as the control. Using the comparative threshold cycle (Ct) method or standard method, the relative quantification of Srx gene expression was calculated after normalization against B2M for each sample. The primers for real-time PCR were designed as follows: Srx primer, forward 5′-CAT CGA TGT CCT CTG GAT CA-3′, and reverse 5′-CTG CAA GTC TGG TGT GGA TG-3′; B2M primer, forward 5′-CTC GCT CCG TGG CCT TAG-3′, and reverse 5′-CAA ATG CGG CAT CTT CAA-3′.

### 4.5. Protein Isolation and Western Blotting

Cells were lysed in lysis buffer A [20 mM HEPES (pH 7.5), 150 mM NaCl, 1 mM EDTA, 2 mM EGTA, 1% Triton X-100, 10% glycerol, and protease cocktail I/II; Sigma], and cellular debris was removed by centrifugation at 10,000× *g* for 10 min. Proteins were separated by sodium dodecyl sulfate polyacrylamide gel electrophoresis, transferred onto nitrocellulose membranes, blocked with 5% skim milk in 0.01 M TBS (pH 7.5) containing 0.5% Tween 20, and blotted with the appropriate primary antibodies. The antigen–antibody complexes were detected by chemiluminescence (Abclone, Seoul, Korea). All of the uncropped images of western blot used in Figures can be found it Appendix A.

### 4.6. Measurement of Mitochondrial Activity

To detect and measure the mitochondrial membrane potential, ROS generation, and Ca^2+^ concentration, we used the specific fluorescent probes TMRE, MitoSOX, and Rhod-2AM (Invitrogen, Carlsbad, CA, USA), respectively. Cells were cultured and then incubated with 1 μM MitoSOX for 20 min or with 5 μM TMRE or Rhod-2AM for 30 min at 37 °C. Fluorescent probe levels were measured using a FACSCanto II flow cytometer (BD Bioscience).

### 4.7. Clonogenic Assay and Colony-Forming Assay

Cells were plated at equal densities in 6-well plates for 24 h and then treated with frugoside at the doses and times indicated in the Figures. After treatment, the cells were trypsinized, serially diluted, and re-plated. Cells were grown for 10 d, and colonies were fixed and stained with crystal violet. Subsequently, the stained dyes were extracted, and the optical density (OD) at 560 nm was measured by a 96-well plate reader (molecular devices). Anchorage-independent growth was assessed by performing colony-forming assays in soft agar assays. Cells were suspended in a 1 mL cell growth medium containing 0.3% agar and plated over a layer of 0.6% agar in a growth medium. Cells were grown at 37 °C with 5% CO_2_. Fifteen-day post-inoculation, the colonies were stained with 0.01% crystal violet for 10 min and counted.

### 4.8. Subcellular Fractionation

For the mitochondrial leak experiments, the cytosolic and mitochondrial fractions were measured using a ProteoExtract subcellular proteome extraction kit (Sigma-Aldrich, St. Louis, MO, USA). Briefly, melanoma cells (1 × 10^6^) were harvested, rinsed twice with ice-cold PBS, and resuspended in 200 μL of extraction buffer 1 containing a protease inhibitor cocktail (PIC). Subsequently, the cells were incubated with gentle agitation at 4 °C for 10 min. The supernatants were separated into pellets and cytosolic fractions by centrifugation at 1000× *g* for 10 min. The pellets were resuspended in 200 μL of extraction buffer 2 containing PIC, incubated with gentle agitation at 4 °C for 30 min, and then separated into pellets and mitochondrial fractions by centrifugation at 6000× *g* for 10 min.

### 4.9. Evaluation of Tumorigenicity and Toxicity

Tumorigenicity and toxicity was determined by subcutaneously injecting M14 melanoma cells into the flanks of 6-week-old female nude mice (Tokyo, Japan). All mice were maintained in accordance with the institutional guidelines of the University of Ulsan Animal Care Committee. After 14 d, when well-established tumors of approximately 50 mm^3^ were detected, 5 mice per group were intraperitoneally injected with frugoside (100 μg/kg/d, 2 d/wk for 17 d), and control mice were injected with phosphate-buffered saline. The tumor size was measured every 2 d using a digital caliper. Tumor volumes were measured using the formula:
V = a × b^2^/2
(1)
where a and b are the largest and smallest superficial diameters of the tumor. At day 17, the tumor masses extracted from each group of mice were photographed.

### 4.10. Statistical Analysis

Data were analyzed using the Student’s *t* test with SigmaPlot 12.0 software (2013, Systat Software Inc., San Jose, CA, USA). *p* values were derived to assess statistical significance and are indicated as follows: * *p <* 0.05; ** *p* < 0.01; and *** *p* < 0.001. Data for all figures are expressed as the mean ± SD of three independent experiments.

## 5. Conclusions

The present study demonstrated that frugoside is a promising antitumor drug that acts by inducing mitochondria-mediated apoptotic cell death in tumor cells. Frugoside may be beneficial for the prevention of melanoma. The mechanism of Srx protein expression regulation by frugoside will be explored in future studies.

## Figures and Tables

**Figure 1 cancers-11-00854-f001:**
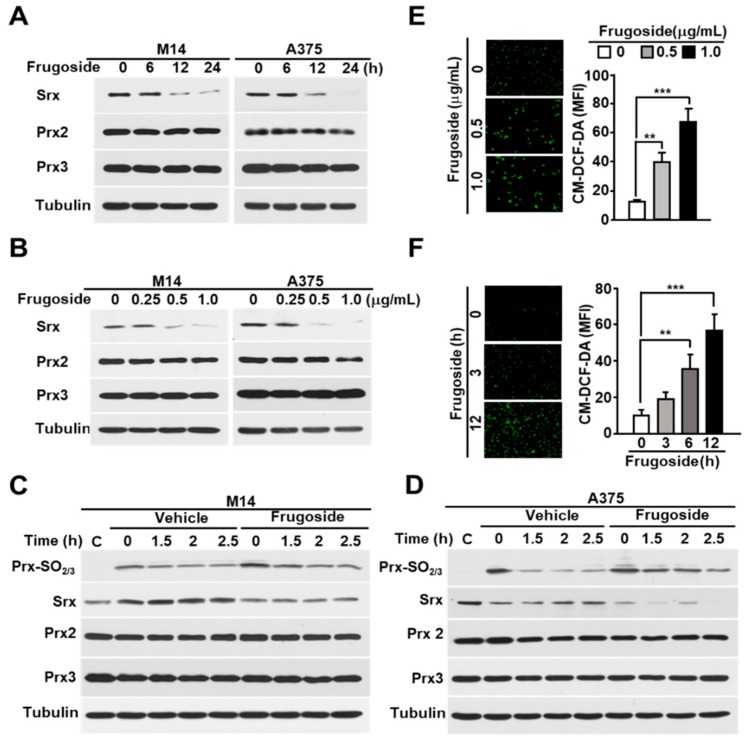
Frugoside inhibits Srx expression, leading to the hyperoxidation of Prxs caused by reactive oxygen species (ROS) accumulation. (**A**,**B**) Melanoma M14 and A375 cells were treated with frugoside in (**A**) time-dependent (0.5 μg/mL) and (**B**) dose-dependent manners for 24 h, and the cells were subjected to a western blot analysis with antibodies against Srx, cytosolic Prx (Prx2), mitochondrial Prx (Prx3), and the loading control, tubulin. (**C**,**D**) M14 (**C**) and A375 (**D**) cells were treated with 200 μM H_2_O_2_ for 10 min with 0.5 μg/mL frugoside. After 10 min, H_2_O_2_ was washed out, and cells were incubated for the indicated times in a fresh medium. As a control of H_2_O_2_-stimulated samples, sample C was harvested without H_2_O_2_ stimulation. The lysates were assessed by a western blot analysis using antibodies against the indicated proteins. (**E**,**F**) M14 cells were treated with different doses of frugoside for 12 h (**E**) or 0.5 μg/mL of frugoside for different lengths of time (**F**). Dichlorofluorescein (DCF) fluorescence was analyzed with a fluorescence microscope after staining with 5 μM CM-H2DCFDA at 37 °C for 30 min. Representative fluorescent microscope images are shown, with quantified data given as a graph in the right panel. Data represent mean ± SD. *p* values were derived to assess statistical significance and are indicated as follows: ** *p* < 0.01; and *** *p* < 0.001. Original magfication, 100×.

**Figure 2 cancers-11-00854-f002:**
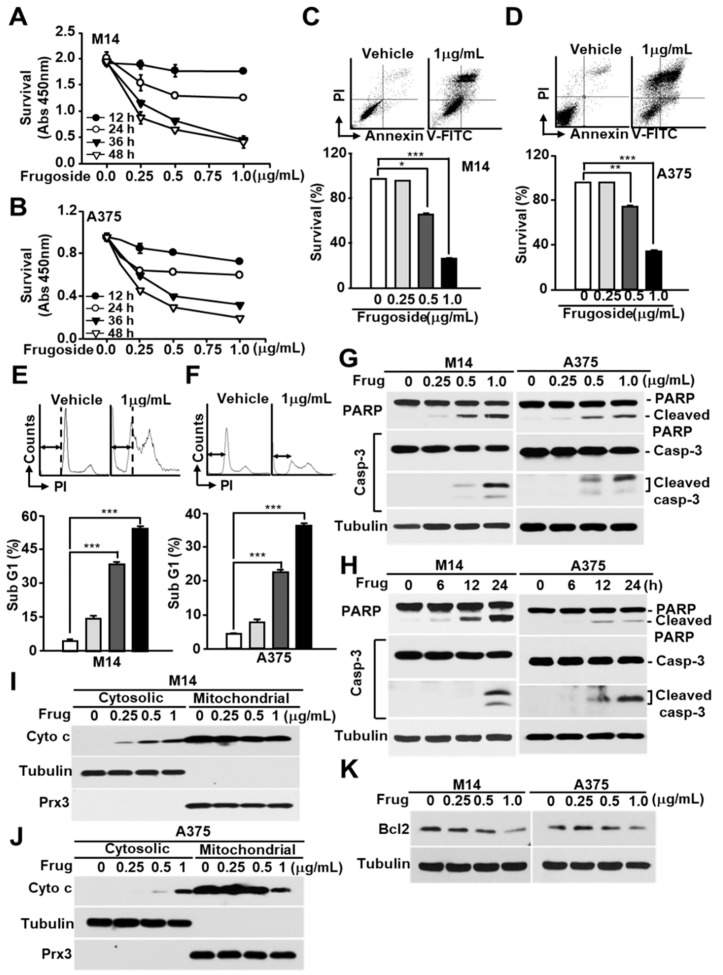
Frugoside induces mitochondria-mediated cell death in melanoma cells. (**A**,**B**) M14 (**A**) and A375 (**B**) cells were treated with frugoside at the indicated time and dose. The cytotoxicity level was measured by the CCK-8 assay. (**C**–**F**) M14 and A375 cells were treated with frugoside at the indicated dose for 24 h. The viability of frugoside-treated M14 (**C**) and A375 (**D**) cells was determined by a fluorescence-activated cell sorting (FACS) analysis after propidium iodide (PI)/Annexin V-fluorescein isothiocyanate (V-FITC) staining. The apoptotic cell population (Sub-G1) was measured by PI staining and a flow cytometry analysis in frugoside-treated M14 (**E**) and A375 (**F**) cells. (**G**,**H**) M14 and A375 cells were treated with various doses of frugoside for 24 h (**G**) or for the indicated periods (**H**; 0.5 μg/mL). The lysates were assessed by a western blot analysis using antibodies against the indicated proteins. (**I**,**J**) M14 (**I**) and A375 (**J**) cells were treated with frugoside in a dose-dependent manner for 24 h. Cytosolic and mitochondrial fractions were prepared from the frugoside treated cells. The identity of the cytosolic and mitochondrial fractions was verified by the specific marker proteins, tubulin and Prx3. (**K**) M14 and A375 cells were treated with frugoside at the indicated doses for 24 h. The cells were analyzed by a western blot analysis with antibodies against Bcl2 and tubulin. *p* values were derived to assess statistical significance and are indicated as follows: * *p <* 0.05; ** *p* < 0.01; and *** *p* < 0.001.

**Figure 3 cancers-11-00854-f003:**
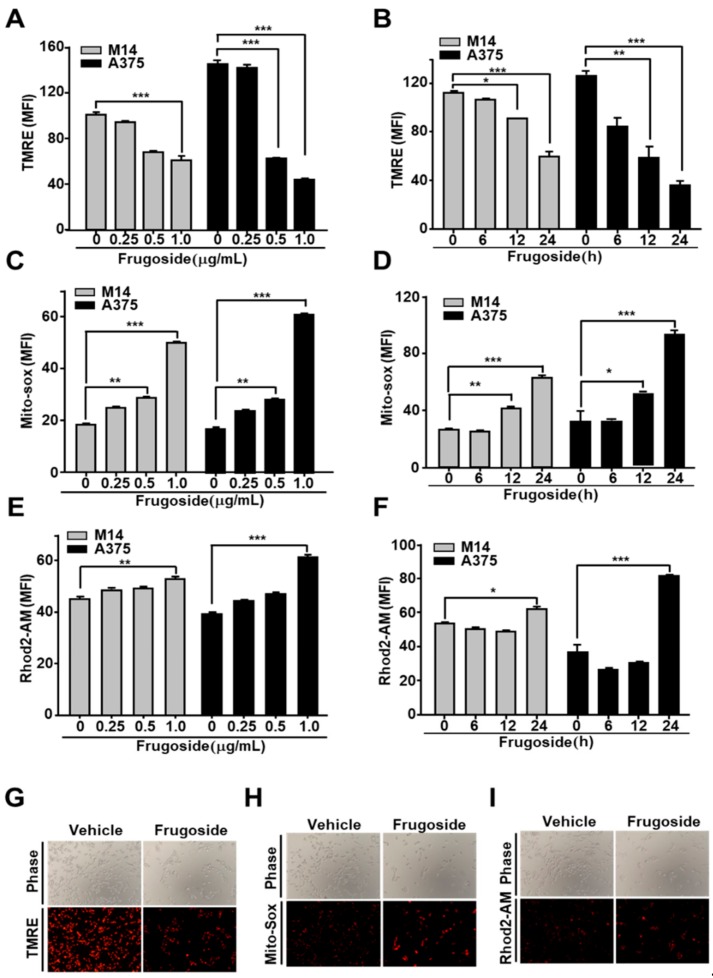
Frugoside results in mitochondrial dysfunction via ROS overproduction. (**A–F**) Melanoma cells were treated with frugoside in dose- (24 h) and time- (0.5 μg/mL dosage) dependent manners. Stimulated cells were analyzed using flow cytometry for mitochondrial conditions, including the membrane potential, mitochondrial ROS, and calcium overload. The mitochondrial membrane potential was measured using a FACSCanto II apparatus after tetraethylrhodamine ethyl ester (TMRE) staining in M14 and A375 cells treated with various doses of frugoside (**A**) over time (**B**). Mitochondrial ROS was measured in frugoside-treated melanoma cells in dose- (**C**) and time- (**D**) dependent manners via MitoSOX staining. The mitochondrial calcium level was measured using Rhod2-AM fluorescent dye and quantified (dose: **E**, time: **F**). Values are presented as mean ± SD. (**G–I**) Melanoma cells were treated with 0.5 μg/mL frugoside for 12 h and then stained with TMRE, MitoSOX, and Rhod2-AM dye. The membrane potential (**G**), mitochondrial ROS (**H**), and mitochondrial calcium level (**I**) were also analyzed using fluorescence microscopy. *p* values were derived to assess statistical significance and are indicated as follows: * *p <* 0.05; ** *p* < 0.01; and *** *p* < 0.001. Original magfication, 100×.

**Figure 4 cancers-11-00854-f004:**
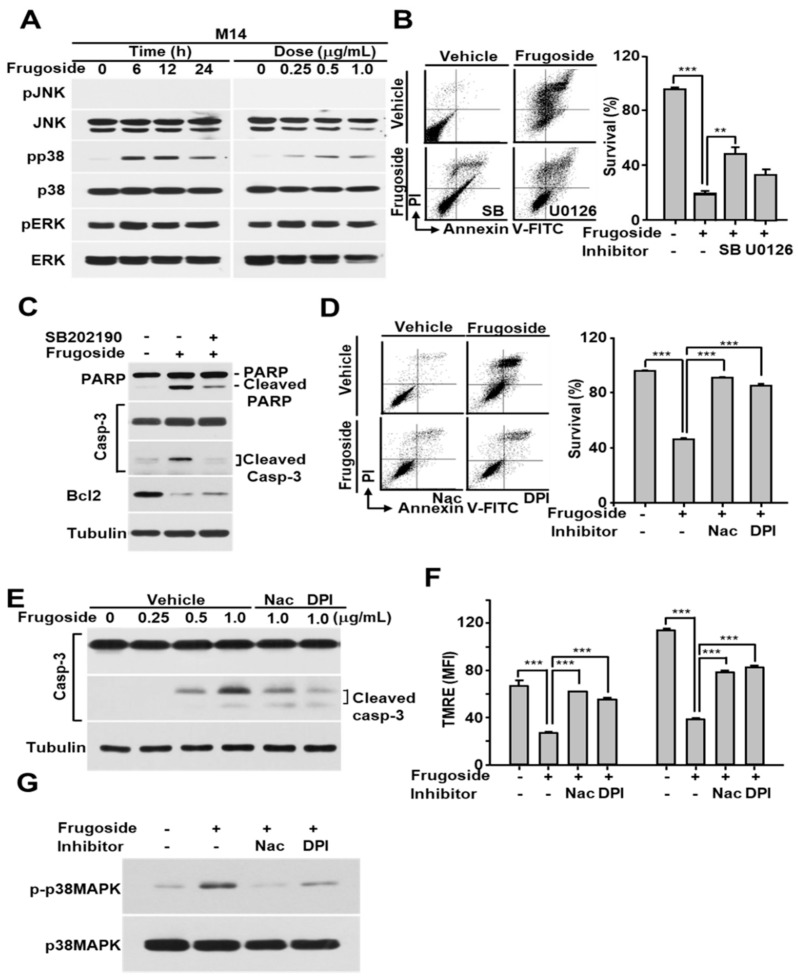
Frugoside-mediated ROS accumulation induces sustained p38 mitogen-activated protein kinase (MAPK) activation, and this activation contributes to apoptotic cell death. (**A**) M14 cells were treated with frugoside in time- and dose-dependent manners as described in Figure 3. The cell lysates were subjected to western blotting with antibodies specific to phosphorylated MAPKs as indicated. The same membrane was re-probed with antibodies to the following MAPKs: c-Jun N-terminal kinases (JNK), p38 MAPK, and extracellular signal-regulated kinase (ERK). (**B**) M14 cells were treated with 0.5 μg/mL frugoside in the presence or absence of the p38 MAPK inhibitor, SB202190, or MAP/ERK kinase (MEK) 1/2 inhibitor, U0126, as indicated. The cells were analyzed for cell death using FACSCanto II after PI/Annexin V-FITC staining. (**C**) M14 cells were treated with 0.5 µg/mL frugoside for 24 h after a 1 h of SB202190 pretreatment. The cell lysates were subjected to western blotting with antibodies against poly (ADP-ribose) polymerase (PARP), caspase-3, Bcl2, and tubulin. (**D**) M14 cells were pretreated with 2 mM N-acetyl cysteine (NAC) or 10 μM diphenyleneiodonium (DPI) for 1 h, followed by 1 μg/mL frugoside for 24 h. The cells were subjected to a FACS analysis after PI/Annexin V-FITC staining. (**E**) M14 cells were treated with frugoside at the indicated doses for 24 h in the presence of 2 mM NAC or 10 μM DPI following a 1 h pretreatment, and the cell lysates were subjected to western blotting with the indicated antibodies. (**F**) M14 cells were pretreated with 2 mM NAC or 10 μM DPI for 1 h, followed by 1 μg/mL frugoside for 24 h, and the cells were analyzed with a FACSCanto II after TMRE staining. (**G**) As described in (**D**), frugoside- and antioxidant-treated M14 cells were subjected to western blotting with antibodies against phosphorylated and non-phosphorylated p38 MAPK. *p* values were derived to assess statistical significance and are indicated as follows: ** *p* < 0.01; and *** *p* < 0.001.

**Figure 5 cancers-11-00854-f005:**
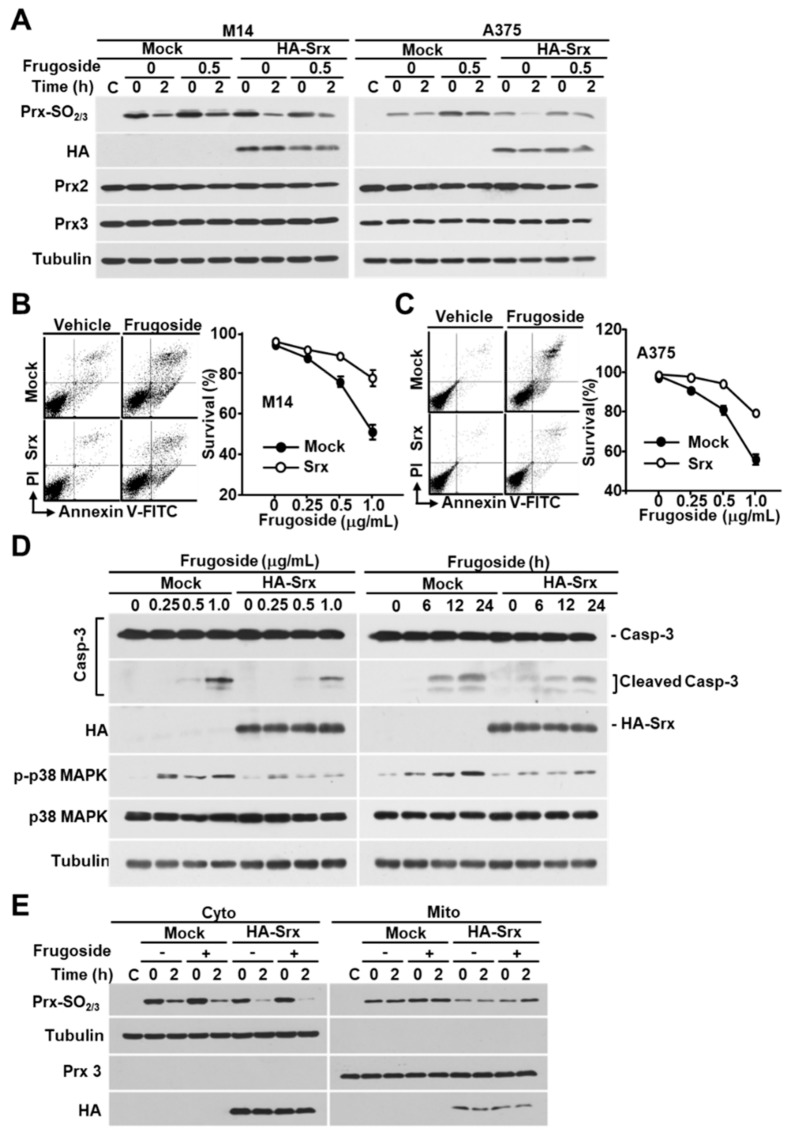
Srx is essential for ROS-mediated cell death by frugoside. (**A**) HA-Srx and Mock plasmids were transfected into M14 and A375 cells. After transfection for 24 h, the cells were treated with or without frugoside (0.5 μg/mL) for 12 h, followed by H_2_O_2_ treatment (200 μM) for 10 min. H_2_O_2_ was removed by replacement with a fresh medium, then the cells were incubated for the indicated times. The lysates were assessed by a western blot analysis using antibodies against the indicated proteins. (**B**,**C**) M14 (**B**) and A375 (**C**) cells were transfected with HA-Srx-overexpressing or Mock plasmids, and the cells were treated with various doses of frugoside. Cell death was measured by a FACS analysis after PI/Annexin V-FITC staining. (**D**) M14 cells were transfected as indicated in (**B**), and the cells were treated with frugoside in dose- and time-dependent manners. The cell lysates were subjected to western blotting with specific antibodies for caspase-3, HA, phospho-p38, p38, and tubulin. (**E**) Srx-overexpressed M14 cells were treated with frugoside and H_2_O_2_, as described in (**A**), and the cells were separated into cytosolic and mitochondrial fractions. The fractionated lysates were subjected to western blotting with antibodies specific to Prx-SO_2_, Prx3, and tubulin.

**Figure 6 cancers-11-00854-f006:**
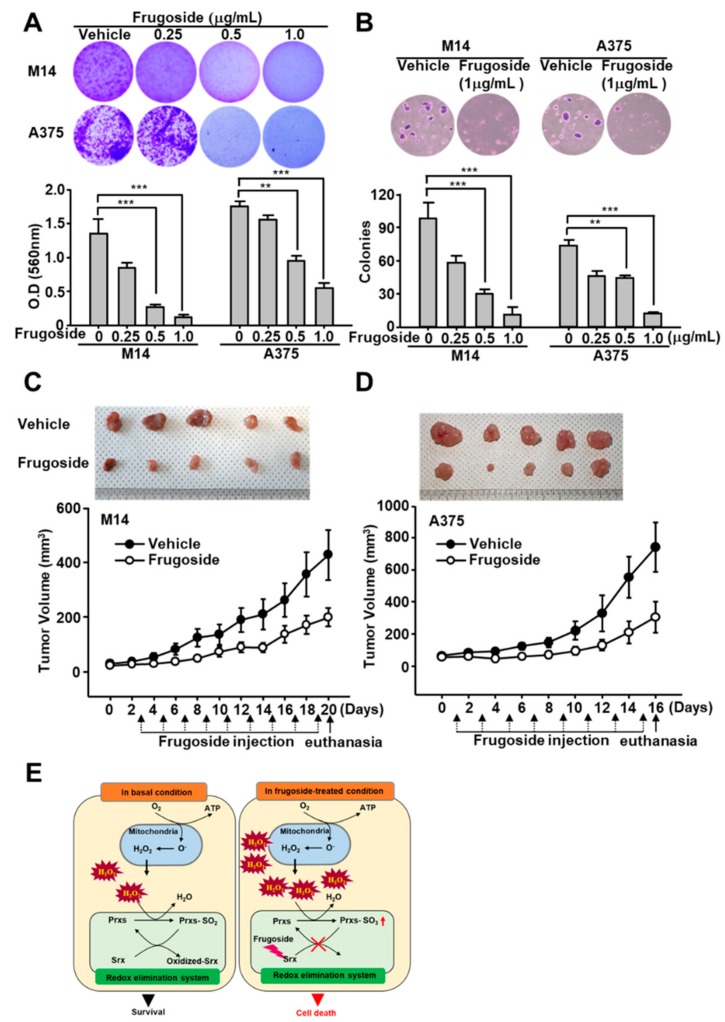
Frugoside inhibits tumorigenic ability in vitro and in vivo. (**A**) A total of 6 × 10^4^ M14 or A375 cells were seeded in a six-well plate. After three days, the cells were treated with frugoside at the indicated dose for one week. The cells were stained with crystal violet solution. Original magfication, 100×. (**B**) M14 and A375 cells were subjected to soft agar assay after 72 h in the medium containing frugoside at the indicated dose, which was changed every five days. The number of colonies generated per 10,000 cells was counted three weeks later. Original magfication, 100×. (**C**,**D**) Frugoside reduced the tumor volume in M14 and A375-inoculated nude mice. To generate the xenograft model, six-week-old BALB/c nude mice were inoculated with 1 × 10^6^ M14 (**C**) and A375 (**D**) cells. After two weeks, frugoside (100 µg/kg in 10% DMSO and 90% PBS) or vehicle were injected once every two days for 17 days. The M14 and A375-inoculated nude mice were sacrificed, and tumor tissue was collected. The photograph shows tumor masses extracted from each group of mice after 17 days (upper panel). (**E**) Schematic illustration: The anti-cancer mechanism of frugoside.

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
