# Peer review of "Frugoside Induces Mitochondria-Mediated Apoptotic Cell Death through Inhibition of Sulfiredoxin Expression in Melanoma Cells"

_cancers, 2019, doi:10.3390/cancers11060854_

Round 1
Reviewer 1 Report
minor: statistical significance should be added in the legend of supplementary Figure 8
Author Response
1. The referee is concerned about the statistical analysis.
Response to Comment:
-As suggested, we have included requested statistical significance in the legend of supplementary Figure 8.
Reviewer 2 Report
The comments raised by me were responded well.
Author Response
He/she seems satisfied with our revision.
Reviewer 3 Report
Though authors have shortened the discussion, yet they have introduced some more data. I feel the data have been cramped. I still do not understand, why they should provide voluminous data.
Author Response
Reviewer#3
Though authors have shortened the discussion, yet they have introduced some more data. I feel the data have been cramped. I still do not understand, why they should provide voluminous data.
Response to Comment:
- As suggested, we tried to reduce the figures to your suggestion. However, reviwer1 and 2 requested more results. So we want you to understand the current situation.
This manuscript is a resubmission of an earlier submission. The following is a list of the peer review reports and author responses from that submission.
Round 1
Reviewer 1 Report
Comments:
The authors found that frugoside inhibited the activity of peroxiredoxins by downregulating sulfiredoxin expression. Resulting accumulation of ROS and stimulated p-p38 activation led to mitochondria-mediated death of two human melanoma cell lines. They reverted this phenotype by Srx overexpression or antioxidants stimulation. The effect on tumor growth was confirmed in xenograft assays.
This is an interesting work bringing new evidences on the potential effect and mechanism of action of natural compounds on the growth of tumor cells. However the low number of tested cell lines (2) weakens the study a lot.
MAJOR comments:
Intro:
P1 L41-42: Mention about the current targeted therapies in melanoma should be at least refered to
p2 L53-54: I don't understand the logic of these sentences. First the authors report the growth
inhibition effect on different cancer lines, but in the next sentence the claim only few reports have been made. They should rather emphasize that no reports were made in melanoma, as they wrote in the abstract
Results:
Most results if not all are shown in M14 and A375. Why choose these lines in
particular since nothing is known regarding anti-cancer functions of frugoside.
why A2058, and LOX-IMVI are shown in supplemental ?
It would be critical to at least include a third line for every phenotype shown: increase in ROS, cell death, reversion with overexpression.
In vivo xenografts should be reproduced in at least another cell line. (why is A375 not shown ?)
Fig1C-D: what represents the sample C and what is the difference with the samples Vehicle 0h
or Frugoside 0h ? If similar, why is the expression of Prx-SO23 different? If the samples are different, it should be explained why in the text.
p4 L127: "...frugoside induces mitochondria-mediated apoptotic cell death via the inhibition of Srx expression" : this sentence is not correct because the figure 2 does not prove Srx is invovled in cell death.
Fig 4: specificity of SB should be assessed by showing the regulation of its downstream targets involved in apoptosis. (ie BID)
Fig 7: in vivo experiments: tail injection for lung metastasis is an easy assay to confirm the role of frugoside on melanoma metastatic progression in vivo
Fig7E: in the model the first square is titled "physiological conditions". If the authors talk about the melanoma tumor system, then the word “physiological” may be inappropriate.
Discussion:
There is no mention from the litterature on the numerous melanoma mouse models
(BRAF, NRAS etc...) and the potentially already known aspects regarding
Metabolism/ anti-oxydant effect or the involvement of p38 pathway in melanoma
This should be discussed
L313-314: too vague, please emphasize with examples from the references.
what are therefore the current applications of antioxydants in cancer?
L326-327: Srx expression analysis should be reinforce a lot by investigating the publically available TCGA database (about 400 melanoma patient-derived tumors)
L337-338: The authors intent to provide a hint in the mechanism of action of their compound
by investigating p38. This seems interesting; however, I do not understand the sentence. It should be reformulated.
If the wish is to provide evidence that p38 is involved in frugoside-induced cell death then they should confirm it at least by showing the regulation of a p38 target which is involved in apoptosis (ie BID). This would also show the specificity of the p38 inhibitor used.
Conclusion:
Authors claim the compound is "especially active against melanoma". which other cancer lines have been tested and on what basis is this sentence build?
Minor comments:
Format of the reference numbering is not correct
Have all 4 cell lines been authenticated?
Author Response
29/05/2019
Samuel C. Mok, PhD
Editor-in-Chief
Cancers
Dear Dr. Mok:
We thank the referees for their careful review and providing constructive criticism concerning our manuscript. We have heeded their comments, and have made the necessary corrections and added the revised contents. Our detailed responses follow this letter.
These changes have substantially improved the manuscript that now clearly shows the effects of frugoside on melanoma. Consequently, we ask that you consider the enclosed revised manuscript for publication.
Thank you for your consideration. I look forward to hearing from you.
Sincerely,
Sung-Wuk Jang, Ph.D.

Reviewer 2 Report
The authors described that "there are few reports about the potential anti-cancer activity of frugoside". However, the bioactive compound frugoside, a new cardenolide glycoside, was recently reported to inhibit the growth of various human cancer cells. It was focused on the novel activity of frugoside on speific cancer cells.
Instead of M14 xenograft animal model, in order to investigate the precise mechnism of frugoside, it is required to input more data in mitochondria-mediated death in vitro cells in detail.
ex) Detection of bax/bcl-2 expression levels are required such as in Fig. 2.
The authors focused on ulfiredoxin (Srx) which is an important enzyme that protects against oxidative damage of host cells. It is better to add other enzymes that protects against oxidative damage of host cells in the introduction section. Other antioxidant proteins, such as
Prx2 and Prx3, were described in the result section.
Minor editing of English language and style required:
increased according to frugoside dosage and treatment time in melanoma -->
increased in a dose and time response manner in frugoside treated melanoma
The source and information of compound are needed. The extraction and identification methods for this compound should be described or cited from the article if available.
Author Response

(The authors gave the same response as above.)

Reviewer 3 Report
Concerns:
1) Authors have given voluminous data. Actually it is information overloaded. They could have easily made into 2 or 3 papers. I do not understand the reason why authors have loaded the paper with voluminous data. Actually, after 4 or 5 figures, one get bored and lose track of the paper. It is information overloaded. Authors can try to give data in a manageable form.
2) Minor revision is required for spelling and grammatical style, e.g
a) Line -43 therapies failed to show a significant benefits in melanoma patients
b) Line -80 that s should go to previous line (79)
c) Line 85 – As shown in figures 1A and 1B
d) Line 111 whether frugoside induced cell death by
Author Response

(The authors gave the same response as above.)
